# Investigating Items to Improve the Validity of the Five-Item Healthy Eating Score Compared with the 2015 Healthy Eating Index in a Military Population

**DOI:** 10.3390/nu11020251

**Published:** 2019-01-23

**Authors:** Marissa M. Shams-White, Kenneth Chui, Patricia A. Deuster, Nicola M. McKeown, Aviva Must

**Affiliations:** 1Friedman School of Nutrition Science and Policy, Tufts University, 150 Harrison Avenue, Boston, MA 02111, USA; nicola.mckeown@tufts.edu (N.M.M.); aviva.must@tufts.edu (A.M.); 2Cancer Prevention Fellowship Program, Division of Cancer Prevention, National Cancer Institute, National Institutes of Health 9609 Medical Center Drive, Rockville, MD 20892, USA; 3Department of Public Health and Community Medicine, School of Medicine, Tufts University, 136 Harrison Avenue, Boston, MA 02111, USA; kenneth.chui@tufts.edu; 4Consortium for Health and Military Performance, A DoD Center of Excellence, Department of Military and Emergency Medicine, Uniformed Services University, 4301 Jones Bridge Road, Bethesda, MD 20814, USA; patricia.deuster@usuhs.edu; 5Jean Mayer USDA Human Nutrition Research Center on Aging at Tufts University, 711 Washington Street, Boston, MA 02111, USA

**Keywords:** 2015 Healthy Eating Index, diet quality, dietary assessment tool, Global Assessment Tool, healthy eating score, indices of diet quality, military, nutrition

## Abstract

Military researchers utilize a five-item healthy eating score (HES-5) in the Global Assessment Tool (GAT) questionnaire to quickly assess the overall diet quality of military personnel. This study aimed to modify the HES-5 to improve its validity relative to the 2015 Healthy Eating Index (HEI-2015) in active duty military personnel (*n* = 333). A food frequency questionnaire was used to calculate HEI-2015 scores and to assess sugar-sweetened beverage (SSB) intake in 8-oz (SSB-8) and 12-oz servings. GAT nutrition questions were used to calculate HES-5 scores and capture breakfast and post-exercise recovery fueling snack (RFsnack) frequencies. Two scoring options were considered for the highest RFsnack category: “4” vs. “5” (RFsnack-5). Potential candidates were added alone and in combination to the HES-5 and compared to the HEI-2015 with a Pearson correlation coefficient. Scores with the highest correlations were compared via a *z*-score equation to identify the simplest modification to the HES-5. Correlations between HES-5 and HEI-2015 scores in total participants, males, and females were 0.41, 0.45 and 0.32, respectively. Correlations were most significantly improved in total participants by adding RFsnack-5, SSB-8, RFsnack-5 + SSB-8, and RFsnack-5 + SSB-8 + breakfast, though the addition of SSB-8 + RFsnack-5 performed best (*r* = 0.53). Future work should consider scoring mechanisms, serving sizes, and question wording.

## 1. Introduction

An estimated 63.6% of all active duty U.S. military are overweight or obese [1]. In addition to its associated chronic disease burden and costs, obesity can negatively affect military personnel’s operational readiness (e.g., increase their likelihood of combat injury or mortality) and jeopardize the long-term welfare of the US military and, thereby, our defense [2,3,4,5,6]. Diet can play a major role in reducing obesity risk and promoting the health and physical readiness of military personnel. However, results from the 2011 Department of Defense (DoD) Survey of Health-Related Behaviors Among Military Personnel (HRBS) indicated only 11.2%, 12.9% and 12.7% of military personnel met the objectives for fruit, vegetable and whole grain intake, respectively [7].

Although the 2011 HRBS results indicate a gap between nutritional recommendations and what military personnel generally consume, the nutritional determinants associated with being overweight or obese are still unclear. Findings across studies examining the associations between individual macronutrients and adiposity are inconsistent [8]. Because individuals commonly eat foods together in meals rather than as nutrients or as single foods, a focus on dietary patterns may better capture the complexity of the diet, account for various nutrient interactions, and provide a clearer picture of the diet’s association with adiposity [9].

Dietary patterns reflect a broader look at the various types, quantities, and/or combinations of different foods, beverages, and nutrients in diets, and the frequency with which they are typically consumed [10,11]. One common method used to summarize dietary data as dietary patterns is through a hypothesis-oriented approach, where one evaluates data using set criteria and aggregates dietary components as a score or index [12]. The USDA 2015 Healthy Eating Index (HEI-2015) takes this approach to assess the extent that an individual’s diet follows key recommendations in the 2015–2020 Dietary Guidelines for Americans (DGA) [13]. This approach relies on data from a detailed quantitative or semi-quantitative dietary assessment instrument, such as a food frequency questionnaire (FFQ) or 24-h recall (24HR), to characterize total diet. Though the HEI-2015 was only recently released, the earlier HEI-2010 has been used extensively in research to capture diet quality in relation to the 2010 DGA [14,15]. However, this comprehensive approach to dietary data collection may be impractical in a military context, clinically, or for administration in large studies due to the cost and time burden on participants. Methods to accurately and rapidly measure the diets of personnel are needed.

To address the need for an abbreviated dietary assessment tool, military researchers developed the five-item Healthy Eating Score (HES-5). The HES-5 is a modified version of the USDA HEI-2005 to quickly assess the overall diet; it looks at five dietary components via five questions (intake of fruit, vegetables, whole grains, dairy, and fish) as part of the updated Comprehensive Soldier and Family Fitness (CSF2) Global Assessment Tool (GAT) [7,16]. Though the HES-5 is strongly associated with health-promoting nutrition behaviors, its correlation with dietary quality is based on the HEI-2005 developed relative to the 2005 DGA. It has not been updated and examined with the most recent HEI-2015, which reflects the 2015-2020 DGA [7,16].

The aim of this study was to determine whether modifications to the GAT HES-5 could improve its validity compared with the HEI-2015 and, thus, more accurately assess the most current nutrient DGA [15]. We hypothesized that modifying the HES-5 by adding items to address frequency of (1) breakfast consumption, (2) post-exercise recovery fueling snack (RFsnack) consumption, and (3) sugar-sweetened beverage (SSB) consumption, alone or in combination, would strengthen its correlation with the HEI-2015. These additions may provide a more accurate tool to assess optimal nutrition than is currently available for future military researchers and clinicians.

## 2. Materials and Methods

This study is a secondary analysis of cross-sectional data from the Consortium of Health and Military Performance’s (CHAMP’s) CSF2 Study. Briefly, the study included 519 active duty military personnel to assess their physical fitness and nutritional status. Volunteers were recruited from June 2014 through November 2017 from Fort Bliss (TX) (*n* = 301), Fort Bragg (NC) (*n* = 104), Fort Myer (VA) (*n* = 78), Fort Detrick (*n* = 32), and Naval Support Activity Bethesda (MD) (*n* = 4). All participants provided written consent prior to participation in the study. Of the 519 participants included in CHAMP’s CSF study, 333 were eligible for inclusion in this study. Participants were excluded if they had incomplete dietary data (*n* = 174) or, as proposed by Willett, if they reported energy intake values <800 kcal/day for males and <500 kcal/day for females (*n* = 12) [17,18]. Because this military population may be more active than the civilian population and require higher caloric intakes, Willett’s upper limit cut-offs of >4000 kcal for men and >3500 kcal for women were not applied. The main reason for incomplete dietary data was that participants did not complete the online GAT (i.e., they completed the FFQs in-person but did not complete the GAT online off-site). Approximately 74% of included participants were male (*n* = 247). The study was approved by the Uniformed Service University (USU) and Tufts University Institutional Review Boards.

### 2.1. Dietary Assessment

Dietary data were collected from subjects through an FFQ in-person and the GAT off-site. A semi-quantitative 110-food item Block FFQ was administered via paper and pencil. Subjects specified the quantity of each item typically consumed over the past 3 months, aided by pictures provided to assist in portion size estimations. To represent usual SSB intake, data were collected on the number of times/week each SSB was typically consumed and the number of cans, bottles, or glasses consumed at each time. Responses from the FFQ on frequency and quantity of each SSB item were used to calculate total SSB g/day for each participant. Because serving sizes varied by SSB item on the FFQ (e.g., 8-oz glasses vs. 12-oz cans), SSB g/day were used to calculate total SSB servings/day using both 8-oz (SSB-8) and 12-oz (SSB-12) servings. Both serving size schemes were assessed as potential additions to the HES-5.

Dietary and behavior data were also collected as part of the GAT, an online tool used to assess subjects’ emotional, social, family, spiritual and physical dimensions of well-being [7]. The web-based questionnaire was developed using FluidSurvey software and hosted by USU. Participants either completed the GAT on-site or, due to time restrictions, were given links and numerical identifiers to enter off-site at a computer with Internet access. The questionnaire was voluntary and anonymous (i.e., no identifying information, including names, email addresses, email domains, or IP addresses, were collected.) The physical dimension included the HES-5, which comprise the fruit, vegetable, whole grain, dairy and fish consumption items in the physical dimension. The physical dimension of the GAT also assessed dietary behavior over the past 30 days with single questions on frequency of breakfast consumption (“How many times per week do you eat breakfast?”) and RFsnack consumption within one hour post-exercise (“Do you typically consume a healthy snack within 60 min after a strenuous exercise session? (Examples include 1 piece of fruit, a handful of nuts, 1 small yogurt container, 1 cup of milk, 1 granola bar, or 1 sports bar)”).

### 2.2. Statistical Analysis

Per a data use agreement, the USU research team that conducted the CHAMP’s CSF2 Study transferred de-identified, coded study data electronically via secure, password-protected folders to the Tufts University team conducting the analyses.

All analyses were conducted on the total sample and stratified by sex. FFQ and GAT data were used to calculate the HEI-2015 and HES-5 scores, respectively. The HEI-2015 scores a participant’s diet based on 13 components on a scale from 0 to 100, with a higher score representing greater adherence to DGA and better diet quality [19]. HES-5 scores were calculated from the aforementioned five dietary components to score the quality of a participant’s diet on a scale from 0 to 25. The components and scoring rubrics of the HEI-2015 and HES-5 are detailed in Table 1.

Next, we analyzed responses on breakfast and RFsnack items from the GAT, and SSB consumption from the FFQ, as potential new components to the HES-5 (i.e., candidate HES-5+). The response categories used for each potential component ranged from 0 to 5. Notably, for RFsnack, there were only five response categories compared to the six response categories for breakfast and the HES-5 components. Two different scoring schemes were examined, with the highest category receiving a score of four (RFsnack-4) or five (RFsnack-5). As previously mentioned, SSB was analyzed as both SSB-8 and SSB-12.

All HEI-2015, HES-5, and candidate HES-5+ scores were plotted to assess normality. Given that their distributions were normal, mean (SD) scores were calculated and correlations between scores examined with a parametric Pearson r coefficient. With the correlation between HES-5 and HEI-2015 as a starting point, we added potential candidates one (HES-6), two (HES-7), and three (HES-8) at a time to the HES-5 to assess whether they increased its correlation with the HEI-2015.

Once the strongest candidates were identified, the items with the highest Pearson r coefficients were formally compared to the HES-5 and HEI-2015. The significant differences between their correlations were examined using a *z*-score equation proposed by Meng et al. [20]. Lastly, the goal was to identify the most significant yet simplest modifications to the HES-5. If more than one candidate HES-5+ was significantly stronger than the HES-5 (e.g., both a six-item and seven-item HES), they were further compared with the Meng et al. [20] equation to identify if they significantly differed from one another. All statistical tests were two-sided, with a significance level of 0.05 and conducted using SAS (version 9.4; SAS Institute, Cary, NC, USA).

## 3. Results

The mean HEI-2015 and HES-5 scores for all participants were 61.5 (SD: 10.2) and 14.6 (SD: 5.1), respectively. Mean scores were higher in females than in males (HEI-2015 mean (SD): 65.0 (11.1) vs. 60.3 (9.7), respectively; HES-5 mean (SD): 15.2 (5.2) vs.14.4 (5.0), respectively). As a comparison, the HES-2005 mean score was 68.0 (SD: 10.7) in total participants, 67.1 (SD: 10.1) in males, and 70.4 (SD: 11.9) in females.

The correlations between the HEI-2015 and all variations of the HES-5 are included in Table 2. The correlations between the HES-5 and HEI-2015 in total participants, males, and females were 0.41, 0.45, and 0.32, respectively. Among the HES-6 options, the correlations were slightly stronger when adding RFsnack-5 or SSB-8 than RFsnack-4 or SSB-12, respectively. Thus, all remaining candidate HES-5+ correlations reported in Table 2 and for the remainder of this paper utilize RFsnack-5 and SSB-8. Both the HES-6 (+SSB-8) and HES-6 (+RFsnack-5) correlations with the HEI-2015 were stronger than those seen with the HES-6 (+breakfast). Among the HES-7 options, the addition of SSB-8 and RFsnack-5 together was strongest. The highest correlations overall were seen with the HES-8 (+breakfast, SSB-8, and RFsnack-5) (Table 2).

The candidate HES-5+ with the strongest correlations (i.e., HES-6 (+SSB-8), HES-6 (+RFsnack-5), HES-7 (+SSB-8 and RFsnack-5), and HES-8 (+breakfast, SSB-8, and RFsnack-5)) were formally compared to HES-5 using the Meng equation. All were significantly more highly correlated than the HES-5 with the HEI-2015 (Table 3). Next, they were tested to see if the addition of more items performed better than those with fewer items. The HES-8 was not found to be significantly different from the HES-7 in either males or females (Table 3). Among males, the HES-7 was significantly more highly correlated than the HES-6 (+RFsnack-5) with the HEI-2015, but not significantly different from the HES-6 (+SSB-8). Among females, the HES-7 was significantly more correlated than both HES-6 scores with the HEI-2015 (Table 3).

## 4. Discussion

The HES-5 is designed to assess the overall diet of military service members in relation to the DGA efficiently. We sought to identify items that could strengthen the validity of the HES-5 compared with the HEI-2015 (and thus 2015–2020 DGA), while still limiting the number of questions to keep it as a short, quick dietary assessment tool. Our results suggest that all candidate HES-5+ performed better than the HES-5. However, the HES-6 (+SSB-8) was the best option among males and the HES-7 (+SSB-8 and RFsnack-5) was the best option among females to replace the HES-5. Given that the GAT is administered to both males and females, our findings suggest that the HES-7 (+SSB-8 and RFsnack-5) is preferable to the HES-5 due to its stronger correlation with the HEI-2015.

Any questions added to the HES-5 should enhance the overall picture of an individual’s diet and dietary behaviors. A previous 2013 study by Purvis et al. used bivariate analyses to examine the association between dietary behaviors and HES-5 scores among military service members [7]. Participants in the highest HES-5 quartile were more likely to report eating breakfast six or more times/week, never drinking regular or diet sodas, and a greater frequency of RFsnack patterns [7]. The instrument was also shown to have good internal consistency [7]. These promising preliminary findings guided the decision to focus on breakfast frequency, RFsnack frequency, and SSB consumption as potential candidates for addition to the HES-5. Additionally, breakfast and RFsnack frequency items are already included in the GAT, making them practical and more feasible options to add to the HES-5. SSB was also considered for inclusion because it is a major source of added sugars and associated with obesity [21]. Our findings align with those of Purvis et al.: adding items examining SSB consumption and RFsnack frequency strengthened the correlations between the HES-5 and the HEI-2015. Although the correlation between the HES-6 (+breakfast) with the HEI-2015 was higher than the HES-5, the improvement was not as great as seen with the inclusion of the other two candidate items.

Optimal nutritional fitness is defined by Montain et al. as the availability and consumption of quality food in appropriate quantities and proportions to ensure optimal mission performance and protect against disease and injury [22]. The construct includes three components: diet quality, healthy food choices, and specific nutritional requirements [22]. The accurate assessment of the nutritional fitness of military personnel is important to the military because it can contribute to our understanding of dietary factors’ associations with health outcomes [23]. Purvis et al. found that increased frequency of breakfast and RFsnack were associated with improved physical and cognitive performance, physical fitness and reduced stress, injuries, and anthropometric values [7]. Additionally, they found increased RFsnack was associated with reduced fatigue post-exercise and faster recovery speed, both of which are important to military service members [7]. Research in athletes similarly report that frequent RFsnack consumption immediately post-exercise can aid in replenishing muscle and liver glycogen stores, speed recovery, and have positive benefits on later performance [24,25,26,27]. Items reflecting breakfast frequency and RFsnack dietary behaviors may represent key parts of the diet and health outcomes currently not captured in the HES-5. Conversely, SSB intake is inversely associated with optimal nutrition. It is a major contributor to added sugars in the US adult diet and is positively associated with weight gain and obesity [28,29,30]. Findings from the 2011 HRBS reflect the high levels of SSB consumption in the military: 19.3% of military personnel consumed SSBs two or more times each day [1,21]. The addition of a SSB item thus may strengthen the HES-5 due to its frequency in the diet. However, its consumption may also be representative of overall unfavorable dietary behaviors. Mullie et al. found that SSB consumption was associated with a lower consumption of fruit and vegetables and higher consumption of meat and fast-food; the latter are not currently captured in the HES-5 [29]. One may surmise that an SSB item may reflect these other dietary behaviors. Further research on SSB and dietary patterns is needed to test this hypothesis.

For researchers and clinicians using the HES-5 outside of the GAT, both RFsnack-5 and SSB-8 items may be useful additions to the HES-5. The addition of the RFsnack-5 item to the HES-5 is ideal given that it is currently part of the GAT. Scoring the highest category of the RFsnack item as a five (+RFsnack-5) made it a stronger candidate than a score of four (+RFsnack-4). Results of this study also suggest that an item on SSB may be particularly useful if added to the GAT, as the HES-6 (+SSB-8) performed even more strongly in this group of participants than the HES-6 (+RFsnack-5). Though measuring SSBs in 8-oz servings strengthened the correlation with the HEI-2015 more than the 12-oz servings, this may be related to the fact that SSBs other than soft drinks typically are measured in 8-oz servings. If the focus in future studies is only on sodas, using a 12-oz serving may be preferable.

There are a few limitations to this study. First, although the SSB item performed well, it was based on several FFQ items that use as examples various SSBs that differ in typical serving size. It is unclear if the distinction among these different SSBs captured by the FFQ items would be considered by participants when completing a single composite SSB item. Future research is warranted to identify whether phrasing could aid in accurately capturing the consumption of various SSB items in a single question. This study was also conducted in a relatively small sample of military personnel volunteers and may not be generalizable to the military population. Future studies are needed among additional military populations to confirm these findings.

However, a few strengths of this study are noteworthy. To the best of our knowledge, this is the first study to examine modifications to the HES-5 to improve its correlation with the HEI-2015. Results suggest that it is possible to significantly improve the correlation of the HES-5 with the addition of just two questions: one on SSB intake and one on RFsnack frequency. Additionally, the RFsnack and breakfast items are already currently part of the GAT. Inasmuch as the Army population completes the GAT annually, it should be possible to examine the addition of both items to the HES-5 in future studies to confirm these findings. Lastly, although previous studies in the military focused primarily on males, almost 26% of study participants were female military service members. It is promising that the improvement in scores was observed in both males and females.

## 5. Conclusions

The HES-5 correlated well with the HEI-2015, although not as well as with the HEI-2005. Results from this study suggest that adding items on the consumption of SSBs and frequency of RFsnack (i.e., HES-7) can strengthen the correlation of the HES-5 with the HEI-2015 for both males and females in the military. Scoring mechanisms, serving sizes, and question wording should be considered in future studies to ensure the ideal components are added to the HES-5.

## Figures and Tables

**Table 1 nutrients-11-00251-t001:** HEI-2015 and HES-5 components and scoring rubric ^1,2^.

HEI-2015 ^3^ components		Score range	Standard for maximum score		Standard for minimum score of zero	
**Assessing Diet Adequacy (higher score indicates higher consumption)**						
Total Fruits ^4^		0–5	≥0.8 cup equiv./1000 kcal		No fruit	
Whole Fruits ^5^		0–5	≥0.4 cup equiv./1000 kcal		No whole fruit	
Total Vegetables ^6^		0–5	≥1.1 cup equiv./1000 kcal		No vegetables	
Greens and Beans ^6^		0–5	≥0.2 cup equiv./1000 kcal		No dark-green vegetables or legumes	
Whole Grains		0–10	≥1.5 oz equiv./1000 kcal		No whole grains	
Dairy ^7^		0–10	≥1.3 cup equiv./1000 kcal		No dairy	
Total Protein Foods ^8^		0–5	≥ 2.5 oz equiv./1000 kcal		No protein foods	
Seafood and Plant Proteins ^8,9^		0–5	≥ 0.8 oz equiv./1000 kcal		No seafood or plant proteins	
Fatty Acids ^10^		0–10	(PUFAs + MUFAs)/SFAs > 2.5		(PUFAs + MUFAs)/SFAs ≤ 1.2	
**To be consumed in moderation (higher score indicates lower consumption)**						
Refined Grains		0–10	≤1.8 oz equiv/1000 kcal		≥ 4.3 oz equiv/1000 kcal	
Sodium		0–10	≤1.1 g/1000 kcal		≥ 2.0 g/1000 kcal	
Added Sugars		0–10	≤6.5% of energy		≥ 26% of energy	
Saturated Fats		0–10	≤8% of energy		≥ 16% of energy	
Total HEI-2015:		Maximum score: 100			Minimum score: 0	
**Frequency of consumption and scores ^11^**						
**HES-5 components**	**≥4 serv/d**	**2–3 serv/d**	**1 serv/d**	**3–6 serv/wk**	**1–2 serv/wk**	**Rarely or Never**
Fruits ^5^	5	4	3	2	1	0
Vegetables ^12^	5	4	3	2	1	0
Whole Grains	5	4	3	2	1	0
Dairy ^13^	5	4	3	2	1	0
Fish ^14^	5	5	5	5	3	0
Total HES-5:		Maximum score: 25			Minimum score: 0	

^1^ The HEI-2015 table is adapted from an article the National Cancer Institute’s Epidemiology and Genomics Research Program website [19]. The HES-5 table is adapted from an article by Purvis et al. published in the *US Army Medical Department Journal* [7]. ^2^ D, day; equiv, equivalent; HEI-2015, 2015 Healthy Eating Index, g, grams; HES-5, 5-item healthy eating score; kcal, kilocalories; MUFAs, monounsaturated fatty acids; oz, ounce; PUFAs, polyunsaturated fatty acids; SFAs, saturated fatty acids; serv, servings; wk, week; ^3^ Intakes between the minimum and maximum standards are scored proportionately. ^4^ Includes 100% fruit juice. ^5^ Includes all forms except juice. ^6^ Includes legumes (beans and peas). ^7^ Includes all milk products, such as fluid milk, yogurt, and cheese, and fortified soy beverages. ^8^ Beans and peas are included here (and not with vegetables) when the Total Protein Foods standard is otherwise not met. ^9^ Includes seafood, nuts, seeds, soy products (other than beverages), and legumes (beans and peas). ^10^ Ratio of PUFAs and MUFAs to SFAs; ^11^ The HES-5 asks respondents about the frequency of consumption of the following foods/beverages over the past 30 days. A higher score indicates higher consumption. ^12^ Includes legumes and starchy vegetables. ^13^ Includes all milk products, as well as soy milk or other calcium-fortified foods (e.g., orange juice, breakfast cereals); ^14^ Specifically, tuna, salmon, and non-fried fish.

**Table 2 nutrients-11-00251-t002:** Correlations between HEI-2015 and candidate HES-5+ scores among CHAMP CSF2 Study participants (*n* = 333) ^1^.

	Total (*n* = 333)	Male (*n* = 247)	Female (*n* = 86)
HEI-2015 vs.: HES-5	*r* = 0.41	*r* = 0.45	*r* = 0.32 ^2^
**HES-6**			
+ RFsnack-4	0.45	0.48	0.38 ^3^
**+ RFsnack-5**	**0.46**	**0.48**	**0.39 ^3^**
+ Breakfast	0.44	0.46	0.39 ^3^
**+ SSB-8**	**0.51**	**0.53**	**0.41**
+ SSB-12	0.50	0.53	0.41
**HES-7**			
+ Breakfast and RFsnack-5	0.48	0.49	0.44
**+ SSB-8 and RFsnack-5**	**0.53**	**0.55**	**0.47**
+ Breakfast and SSB-8	0.53	0.54	0.46
**HES-8**			
**+** **Breakfast, RFsnack-5, SSB-8**	**0.55**	**0.56**	**0.50**

^1^ Candidate HES-5+, updated healthy eating score with more than five items; CHAMP, Consortium of Health and Military Performance; CSF2 Study, Comprehensive Soldier and Family Fitness Study; HEI-2015, 2015 Healthy Eating Index; HES-5, 5-item Healthy Eating Score; HES-6, 6-item Healthy Eating Score; HES-7, 7-item Healthy Eating Score, HES-8, 8-item Healthy Eating Score, SSB-8, RFsnack-4, post-exercise recovery fueling snack with scoring option A; RFsnack-5, post-exercise recovery fueling snack with scoring option B; SSB-8, sugar-sweetened beverages with 8-oz/serving; SSB-12, sugar-sweetened beverages with 12 oz/serving. The candidate HES-5+ items with the strongest correlations are bolded. Unless noted, all *p*-values were <0.0001. ^2^
*p* = 0.0003; ^3^
*p* < 0.001.

**Table 3 nutrients-11-00251-t003:** A formal comparison of correlated correlation coefficients among CHAMP’s CSF military participants: HES-5 and HES 5+ vs. HEI-2015 (*n* = 333) ^1^.

	Total (*n* = 333)	Males (*n* = 247)	Females (*n* = 86)
**Comparing candidate HES-5+ to HES-5 vs. HEI-2015:**			
HES-6 (+SSB-8)			
*r*_x_: HES-6 vs. HES-5	0.92	0.92	0.93
*r*_1_: HES-5 vs. HEI-2015	0.41	0.45	0.32
*r*_2_: HES-6 vs. HEI-2015	0.51	0.53	0.41
*z*-score:	−4.94 ^***^	−3.90 ^***^	−2.49 ^*^
HES-6 (+RFsnack-5)			
*r*_x_: HES-6 vs. HES-5	0.97	0.97	0.97
*r*_1_: HES-5 vs. HEI-2015	0.41	0.45	0.32
*r*_2_: HES-6 vs. HEI-2015	0.46	0.48	0.39
*z*-score:	−3.52 ^***^	−2.40 ^*^	−3.27 ^**^
HES-7			
*r*_x_: HES-7 vs. HES-5	0.90	0.89	0.91
*r*_1_: HES-5 vs. HEI-2015	0.41	0.45	0.32
*r*_2_: HES-7 vs. HEI-2015	0.53	0.55	0.47
*z*-score:	−5.62 ^***^	−4.21 ^***^	−3.54 ^***^
HES-8			
*r*_x_: HES-8 vs. HES-5	0.92	0.91	0.92
*r*_1_: HES-5 vs. HEI-2015	0.41	0.45	0.32
*r*_2_: HES-8 vs. HEI-2015	0.51	0.52	0.48
*z*-score:	−5.10 ^***^	−3.33 ^***^	−4.13 ^***^
**Comparing candidate HES-5+ versions vs. HEI-2015:**			
HES-7 vs. HES-6 (+SSB-8)			
*r*_x_: HES-7 vs. HES-6	0.98	0.97	0.98
*r*_1_: HES-6 vs. HEI-2015	0.51	0.53	0.41
*r*_2_: HES-7 vs. HEI-2015	0.53	0.55	0.47
*z*-score	−2.65 ^**^	−1.71	−2.78 ^**^
HES-7 vs. HES-6 (+RFsnack-5)			
*r*_x_: HES-7 vs. HES-6	0.94	0.94	0.95
*r*_1_: HES-6 vs. HEI-2015	0.46	0.48	0.39
*r*_2_: HES-7 vs. HEI-2015	0.53	0.55	0.47
*z*-score	−4.68 ^***^	−3.68 ^***^	−2.29 ^*^
HES-6 (+SSB-8) vs. HES-8			
*r*_x_: HES-8 vs. HES-6	0.91	0.90	0.93
*r*_1_: HES-6 vs. HEI-2015	0.51	0.53	0.41
*r*_2_: HES-8 vs. HEI-2015	0.51	0.52	0.48
*z*-score	−0.41	−0.34	−1.89
HES-6 (+RFsnack-5) vs. HES-8			
*r*_x_: HES-8 vs. HES-6	0.96	0.96	0.96
*r*_1_: HES-6 vs. HEI-2015	0.46	0.48	0.39
*r*_2_: HES-8 vs. HEI-2015	0.51	0.52	0.48
*z*-score	−4.32 ^***^	−2.73 ^**^	−3.12 ^**^
HES-7 vs. HES-8			
*r* _x_	0.95	0.94	0.96
*r* _1_	0.53	0.55	0.47
*r* _2_	0.51	0.52	0.48
*z*-score	−1.29	−1.62	−0.54

^1^ Candidate HES-5+, healthy eating score with additional item(s); CHAMP, Consortium of Health and Military Performance; CSF2 Study, Comprehensive Soldier and Family Fitness Study; HEI-2015, 2015 Healthy Eating Index; HES-5, 5-item healthy eating score; HES-6, 6-item healthy eating score; HES-7, 7-item healthy eating score including SSB-8 and RFsnack-5 items; HES-8, 8-item healthy eating score including breakfast, SSB-8 and RFsnack-5 items; *r*_1_, *r*_2_, and *r*_x_, Pearson *r* coefficients between scores; RFsnack-5, post-exercise recovery fueling snack with scoring option B; SSB-8, sugar-sweetened beverages with 8-oz/serving. All “HES-7” are referring to HES-7(+SSB-8 and RFsnack-5); all HES-8 refer to HES-8(+breakfast, SSB-8 and RFsnack-5). All correlations were used to calculate *z*-scores using equations proposed by Meng [20]. Unless noted, *p* > 0.05; * *p* < 0.05; ** *p* < 0.01; *** *p* < 0.001.

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
