# Peer review of "Investigating Items to Improve the Validity of the Five-Item Healthy Eating Score Compared with the 2015 Healthy Eating Index in a Military Population"

_nutrients, 2019, doi:10.3390/nu11020251_

Reviewer 1 Report

Shams_White and co-workers aimed to improve a 5-item healthy eating score (HES-5) in the Global Assessment Tool (GAT) questionnaire to quickly assess the overall diet quality of military personnel. By modifying the HES-5, they tested the validity relative to the 2015 Healthy Eating Index (HEI-2015) in active duty military personnel. The Authors used  a food frequency questionnaire to calculate HEI-2015 scores and to assess sugar-sweetened beverage (SSB) intake in 8-oz (SSB-8) and 12-oz servings. GAT nutrition questions were used to calculate HES-5 scores and capture breakfast and post-exercise recovery fueling snack (RFsnack) frequencies. Two scoring options were considered for the highest RFsnack category: “4” vs. “5” (RFsnack-5). Potential candidates were added alone and in combination to the HES-5 and compared to the HEI-2015 with a Pearson correlation coefficient. Scores with the highest correlations were compared via a Z-score equation  to identify the simplest modification to the HES-5. Correlations between HES-5 and HEI-2015 scores in total participants, males, and females were 0.41, 0.45 and 0.32, respectively. Correlations were most significantly improved in total participants by adding RFsnack-5, SSB-8, RFsnack-5+SSB-8, and RFsnack-5+SSB-8+breakfast, though the addition of SSB-8+RFsnack-5 performed best (r=0.53). Therefore, they concluded that to identify items could strengthen the validity of the HES-5 223, if compared with the HEI-2015 (and thus 2015-2020 DGA).

The study is of interest, original, and with a high generalizability. It could be an effective approach for improving food intake in different settings. References are appropriates, the statistics is in line with the type of the study.

Author Response

Dear reviewer,

Thank you very much for your thoughtful review of this paper. We greatly appreciate your time and comments.

Best Regards

Reviewer 2 Report

This is a well presented manuscript. The authors have done an excellent job describing HEI-2015, the need for HES-5 and correlation with HEI-2015. Limitations have been adequately acknowledged especially that meat and fast food consumption that was not captured by HES-5. I have a couple of minor comments:

Line 43  - Please specify if the overweight and obesity statistics represent those on active military duty, reserve forces or it encompasses all branches of military.

Line 167: delete "correlated" at the beginning

Author Response

Dear Reviewer,

Point 1: Line 43  - Please specify if the overweight and obesity statistics represent those on active military duty, reserve forces or it encompasses all branches of military.

Response 1: Thank you for your comment. This line is now updated to specify that this statistic represents active duty military as follows: "An estimated 63.6% of all active duty U.S. military are overweight or obese [1]."

Point 2: Line 167: delete "correlated" at the beginning

Response 2: Thank you for your edit, "correlated" is now deleted. 

Thank you for your time and comments, we greatly appreciate it.

Best Regards